# Consequences Matter: Compassion in Conservation Means Caring for Individuals, Populations and Species

**DOI:** 10.3390/ani9121115

**Published:** 2019-12-11

**Authors:** Paul J. Johnson, Vanessa M. Adams, Doug P. Armstrong, Sandra E. Baker, Duan Biggs, Luigi Boitani, Alayne Cotterill, Emma Dale, Holly O’Donnell, David J. T. Douglas, Egil Droge, John G. Ewen, Ruth E. Feber, Piero Genovesi, Clive Hambler, Bart J. Harmsen, Lauren A. Harrington, Amy Hinks, Joelene Hughes, Lydia Katsis, Andrew Loveridge, Axel Moehrenschlager, Christopher O’Kane, Meshach Pierre, Steve Redpath, Lovemore Sibanda, Pritpal Soorae, Mark Stanley Price, Peter Tyrrell, Alexandra Zimmermann, Amy Dickman

**Affiliations:** 1Wildlife Conservation Research Unit, Department of Zoology, University of Oxford, The Recanati-Kaplan Centre, Tubney, Oxfordshire OX13 5QL, UK; paul.johnson@zoo.ox.ac.uk (P.J.J.); sandra.baker@zoo.ox.ac.uk (S.E.B.); alayne.cotterill@gmail.com (A.C.); emma.dale@lmh.ox.ac.uk (E.D.); hollyodonnell146@gmail.com (H.O.); egil.droge@zoo.ox.ac.uk (E.D.); ruth.feber@zoo.ox.ac.uk (R.E.F.); lauren.harrington@zoo.ox.ac.uk (L.A.H.); amy.hinks@zoo.ox.ac.uk (A.H.); LydiaKat@hotmail.co.uk (L.K.); tawnycat1@hotmail.com (A.L.); axelm@calgaryzoo.com (A.M.); christopher.a.j.okane@gmail.com (C.O.); meshachpierre@gmail.com (M.P.); lovemore.sibanda@zoo.ox.ac.uk (L.S.); mark.stanleyprice@zoo.ox.ac.uk (M.S.P.); peter.tyrrell@zoo.ox.ac.uk (P.T.); alexandra.zimmermann@zoo.ox.ac.uk (A.Z.); 2School of Technology, Environments & Design University of Tasmania, Hobart TAS 7005, Australia; vm.adams@utas.edu.au; 3Wildlife Ecology Group, Massey University, Palmerston North 4442, New Zealand; D.P.Armstrong@massey.ac.nz; 4Environmental Futures Research Institute, Griffith University, Nathan Queensland 4111, Australia; d.biggs@griffith.edu.au; 5ARC Centre of Excellence for Environmental Decisions, Centre for Biodiversity & Conservation Science, School of Biological Sciences, University of Queensland, Brisbane, Queensland 4072, Australia; 6Department of Conservation Ecology and Entomology, Stellenbosch University, Private Bag X1, Matieland 7602, South Africa; 7Department Biology and Biotechnologies, University of Rome, Viale Università 32, 00185 Roma, Italy; luigi.boitani@uniroma1.it; 8RSPB Centre for Conservation Science, RSPB Scotland, 2 Lochside View, Edinburgh EH12 9DH, UK; david.douglas@rspb.org.uk; 9Institute of Zoology, Zoological Society of London, Regents Park, London NW1 4RY, UK; John.Ewen@ioz.ac.uk; 10Institute for Environmental Protection and Research and Chair IUCN SSC Invasive Species Specialist Group, Via V. Brancati 48, 00144 Rome, Italy; piero.genovesi@isprambiente.it; 11Department of Zoology, University of Oxford, Mansfield Road, Oxford OX1 3PS, UK; clive.hambler@zoo.ox.ac.uk; 12Panthera, 8 West 40th Street, 18th Floor, New York, NY 10018, USA; bharmsen@panthera.org; 13Government of Belize, Old Lands Building, Market Square, Belmopan, Belize; 14RSPB Centre for Conservation Science, The Lodge, Sandy, Bedfordshire SG19 2DL, UK; joelene.hughes@rspb.org.uk; 15Centre for Conservation Research, Calgary Zoological Society, 1300 Zoo Road NE, Calgary, Alberta, AB T2E 7V6, Canada; 16Zoology Building, University of Aberdeen, Tillydrone Avenue, Aberdeen AB24 2TZ, UK; s.redpath@abdn.ac.uk; 17Environment Agency Abu Dhabi, Al Mamoura Building, Murour Road, Abu Dhabi, UAE; psoorae@gmx.com; 18South Rift Association of Landowners, P.O. Box 15289, Nairobi 00509, Kenya

**Keywords:** ethics, compassion, consequentialism, virtue

## Abstract

**Simple Summary:**

Acting to preserve biodiversity can involve harming individual animals. It has recently been argued that conventional practice has placed too much emphasis on the preservation of collective entities, such as populations and species, at the expense of suffering for individuals. At least some advocates of the ‘Compassionate Conservation’ movement find any deployment of lethal measures in the interests of conservation to be unacceptable. This shifts the balance of priorities too far. While conservationists have a duty to minimise harm, and to use non-lethal measures where feasible, there will be serious implications for conservation if this movement were to be widely influential. Furthermore, the ‘do-no-harm’ maxim the compassionate conservationists advocate does not always promote the welfare of individual animals.

**Abstract:**

Human activity affecting the welfare of wild vertebrates, widely accepted to be sentient, and therefore deserving of moral concern, is widespread. A variety of motives lead to the killing of individual wild animals. These include to provide food, to protect stock and other human interests, and also for sport. The acceptability of such killing is widely believed to vary with the motive and method. Individual vertebrates are also killed by conservationists. Whether securing conservation goals is an adequate reason for such killing has recently been challenged. Conventional conservation practice has tended to prioritise ecological collectives, such as populations and species, when their interests conflict with those of individuals. Supporters of the ‘Compassionate Conservation’ movement argue both that conservationists have neglected animal welfare when such conflicts arise and that no killing for conservation is justified. We counter that conservationists increasingly seek to adhere to high standards of welfare, and that the extreme position advocated by some supporters of ‘Compassionate Conservation’, rooted in virtue ethics, would, if widely accepted, lead to considerable negative effects for conservation. Conservation practice cannot afford to neglect consequences. Moreover, the do-no-harm maxim does not always lead to better outcomes for animal welfare.

## 1. Introduction

Conservation action can have repercussions for animal welfare. Is it ever acceptable to kill individual animals in the interests of protecting a population or a species? May invasive rodents be killed to defend native birds? May problem predators be shot if this promotes tolerance of the predator species by the local people affected? Quandaries like these, impinging on the fate of sentient beings, are not uncommon.

Wallach et al. [1] highlighted the ethical challenges that arise when the interests of individual animals conflict with those of ecological collectives (such as populations or species). They promote a version of ‘compassionate conservation’, placing concern for individuals as a central principle, and argue that the current and historical focus of conservation practice on preserving collectives is not conducive to compassion for individual animals. They claim that conservationists have ‘routinely ignored the impact of action on individuals’ and go so far as to state that conservationists ‘often’ assume a binary choice between compassion for individuals or conservation. This badly misrepresents how conservation is currently practised—conservationists demonstrably and increasingly seek to incorporate high welfare standards into their work. They strive to minimise harm by attending diligently to welfare and by exploring alternatives to lethal control [2,3,4]. There has been a marked increase over the last 20 years in animal welfare science in conservation and wildlife management [5,6]. For example, since early eradication programs targeting invasive pests were thought not to be sufficiently informed by welfare science, considerable work has been done to assess the humaneness of different options [7].

Conservation has not, therefore, ‘largely excluded animal ethics from its moral universe’. Treating animals with some concern for their welfare is indeed not unique to Compassionate Conservation [8]. However, shifting the balance too far in favour of the individual over the collective risks inhibiting our ability to conserve biodiversity, and at a time when the need to act has never been greater [9].

## 2. Conservation: Collectives, Individuals and Ethics

We start by defending the proposition that a focus on collectives is central for any meaningful definition of conservation. In an influential paper, Soulé answered the question ‘What is Conservation Biology?’ by stating that protecting biodiversity was a priority [10]. While conservation as a discipline may have some difficulty in articulating what it aims to achieve [11], this continues to be widely accepted [12]. Wallach et al. acknowledged that tackling ecological damage, including species extinction, is among the greatest challenges faced by humanity.

The authors’ focus on individuals is rooted in their value of compassion as a ‘moral virtue’, part of a system of ‘virtue ethics’, where this and other virtues are manifested by the ‘proper conduct’ of human beings. Contemporary virtue ethics derives from Aristotle and other ancient philosophers who argued that morality principally involves the cultivation of noble character traits [13]. Actions are not necessarily defined as being virtuous by their outcomes but by whether they contribute to that cultivation. One influential account of the modern version of virtue ethics is explicit on this: “It is of the character of a virtue that in order to be effective in producing the internal goods [internal to a human] which are the rewards of the virtue it should be exercised without regard to consequences” [14], p. 198. Later, in the same chapter (p. 199), we find that proper conduct might have a negative impact on non-abstract entities: “…....as we have seen cultivation of the virtues always may and often does hinder those external goods which are the mark of worldly success”. It follows that the ‘external goods’ (such as populations or species) may not only not benefit from the application of virtue ethics motivated by compassion for individual animals, but that harm to them is consistent with virtuous conduct. There are competing virtue theories—Sandler [15] stated that the ‘most prominent’ of these share the ‘crucial feature’ that right action is largely determined by the goods and values in the world, and that character traits can be justified as virtues if they promote environmental goods and services (including ecosystem services). Care for living things and moderation in the use of natural resources are examples, he states, of environmentally justified virtues.

Consequentialism is a different ethical framework and one which has dominated thinking in conservation until recently [11]. It posits that whether an action is judged to be ‘right’ or not is substantially determined by the relative value of its outcomes [11]. This is encapsulated by the well-known maxim: ‘do the ends justify the means?’ Consequentialism (unlike utilitarianism to which it is closely related) is not restricted to comparing outcomes of the same type [16]. From a consequentialist perspective, individual welfare and the conservation outcomes both affect a judgement concerning whether any action is ‘right’ or not. Different types of outcome for individuals and collectives can therefore be weighed against each other in determining whether any action is justified. An obvious criticism of this principle is that, unconstrained, it could condone any action, provided the benefits were sufficiently large [11]. In practice of course, there are constraints, not least in law. Moreover, the divisions between different ethical frameworks are less well defined than often thought—Coals et al. [17], for example, pointed out that ‘end-goals’ can often be expressed as duties and vice versa. The Compassionate Conservation agenda based on virtue ethics arguably leans too hard on rule-based ethics (deontology) at the expense of outcomes. (MacIntyre [14] wrote (p. 244) that ‘a virtue is now generally understood as a disposition or sentiment which will produce in us obedience to certain rules….’). More recently, Sandler [15] pointed to a tendency for convergence among different types of environmental ethics (including virtue ethics and consequentialism) in support of character traits ‘that promote ecological sustainability and against traits that promote ecological degradation’. Some supporters of the compassionate conservation agenda in the form advanced by Wallach et al. have previously advocated pluralism in conservation ethics, rather than adherence to one overarching ethical framework (such as is defended by some prominent ethicists, including, they state, J.B. Callicott) [18]. However, the position advanced in [1] clearly places pre-eminent importance on the virtues and little on consequences.

## 3. The ‘Hard’ Cases

Wallach et al.’s perspective and that of mainstream conservationists diverge in their verdicts concerning what is morally justified behaviour when a conservation goal can be realistically achieved only where harm to individuals results. Prioritising collectives cannot be generalised as leading to unnecessary harm to individuals if a clearly defined conservation goal of sufficient value is achieved. Ethical conservation policy in these cases attends to both collectives and to individual well-being [12,19]. The claim that conservationists neglect the moral status of individuals and ‘often’ assume a binary choice between compassion for individuals and the conservation of collectives is simply wrong. Furthermore, the citation offered in support of this statement [10], and the implication that such a dichotomy is a tenet of conservation biology itself is not appropriate. Soulé [10] does argue that conservation and welfare are conceptually distinct. However, he also states that conservation’s ethical imperative to preserve species ‘does not in any way detract from ethical systems that provide behavioral guidance for humans on appropriate relationships with other species, especially when the callous behaviour of humans causes animals to suffer unnecessarily’.

Wallach et al. alluded to the Kantian maxim that it is wrong to treat people as merely means, by implication extending it to individual animals, which are also widely attributed intrinsic value. Parfit (2011, p. 212) drew attention to the importance of the word ‘merely’ in this maxim, contrasting a hypothetical scientist who experiments on animals with a focus only on a research goal regardless of animal suffering with a second scientist pursuing the same aim but at the same time striving to minimise animal suffering [20]. Only the first is treating the animals merely as a means. In the same way, conservationists who minimise the impact of their action on individuals are clearly not treating them merely as a means to their conservation end.

Acknowledging the intrinsic value of sentient individuals does not preclude that those individuals could also have a use value for humans. While belief in intrinsic value is widespread [21], humans have used animals for various purposes across the globe for millennia [12,22]. Wallach et al. acknowledge that use values can be used to promote conservation but then argue that promoting an instrumentalist stance risks alienating ‘large sectors of the public’. This neglects that ‘publics’ around the globe are likely to differ widely in their attitudes to ecologically sustainable use of wildlife. Many people suffer real costs from wildlife. There are many instances where it is the instrumental value of wildlife that, at least in the short term, provides the only feasible basis for garnering support for its conservation, particularly where people live alongside wildlife that threatens their livelihoods, or even their lives [23].

## 4. Risks of Perverse Outcomes

Wallach et al. argued that doing nothing is preferable to risking ‘causing more harm than good’. The ‘do no harm’ (to individuals) maxim they promote, adapted from medical ethics, does not unequivocally privilege inaction over action in conservation policy: the consequences of inaction may be no easier to predict, or more palatable, than those of intervention. Where invasive species are concerned, passively accepting resulting novel ecosystems deserves appraisal [24]. However, very often, the resulting ‘novel’ ecosystem will be substantially impoverished if no action is taken. Hambler and Canney [12] (p. 305) advocated a case-by-case approach bearing in mind extinction risks and the conservation value of the species concerned.

The harm that results from inaction may not be confined to effects on the environment. Hampton et al. [25] explored the welfare implications of lethal control of overabundant wild herbivores in the broader interests of nature conservation, using an explicitly consequentialist methodology. They showed that lethal control can plausibly lead to a net positive effect on individual wellbeing compared with taking no action. Similarly, Russell et al. [26], Driscoll and Watson [27] and Hayward et al. [6] pointed to lose-lose outcomes for welfare and conservation where lethal control of invasive species is eschewed. Gough island in the South Atlantic is a good example: nearly two million sea-bird chicks are currently predated annually by non-native invasive mice introduced by people, driving some endemic species to the brink of extinction [28]. Direct observation strongly suggests severe suffering for many of the individuals affected [29]. Callen et al. [30] provided other vivid examples of where a ‘do nothing’ policy can lead. Failure to cull feral horses in Australia, for example, has resulted in the starvation of thousands of individual horses (as well as environmental damage). Poisoning by the invasive cane toad (*Rhinella marina*) leads to the death of many predators in more than 50 countries globally—but compassionate conservationists oppose the use of novel methods for their control or eradication. In human-animal conflicts, failure to eliminate problem animals can have lethal consequences both for other animals and for people [31]. While it has been said that these apparently perverse outcomes are problematical for compassionate conservationists and ‘the suffering they seek to prevent by adhering to virtue ethics’ [27], recall that virtue ethics is not necessarily driven primarily by concern for consequences.

## 5. Areas of Agreement

Our perspective and that of Wallach et al. converge in deprecating some interventions. There have undoubtedly been examples of conservation action compromising animal welfare without sufficient reason. Harm to individuals where objectives are unclear or have no realistic prospect of being achieved cannot be justified. Nor can action which is motivated solely by individuals’ status as non-native species, or nominally as ‘pests’, or ‘vermin’ [32]. Similarly, action to achieve a conservation goal which is framed solely as a target number of culled individuals is not justified, as is said to be the case for the ‘War on cats, Australia’ program (cited in Table 1 of [1] as an example of lack of compassion). Such a target should be closely linked to how it benefits conservation [32]. It is also clear that any intervention that entails harm to individuals should go hand-in-hand with efforts to address the anthropogenic root causes of the conservation problem motivating the intervention (in the interests of both welfare and conservation).

We do not dispute that it is possible for an intervention to benefit collectives without harming individuals, or that it is possible for badly devised action to harm both individuals and collectives. Cases where the evidence supports such clear-cut conclusions are ethically unproblematic. There can be no disagreement on supporting the former and condemning the latter. However, even here, apparent success stories may be more complicated than they first appear. For example, Potgieter et al. [33] explored a scenario where guard dogs were deployed to protect stock as an alternative to lethal control of the predators. They found that dogs and farmers then killed more individuals of some predator species than they did before the intervention.

## 6. Conclusions

Many problems, of course, are complex, with no obvious solution. The version of Compassionate Conservation offered in [1] has little to offer where non-lethal options are not feasible, but where the need to act is pressing. The senior author of [1] has confirmed that she finds all killing for conservation to be unacceptable [34]. This represents a clear shift compared with earlier versions of the compassionate conservation movement. In 2012 [35], Draper and Bekoff argued that conservationists should refuse to take any action harming individuals if the objectives were not sufficiently valuable—the clear corollary of this is that such action would be acceptable to those authors provided the gains were sufficiently large. Furthermore, in 2015, Ramp and Bekoff [36] advocated compassionate conservation as a means to provide solutions to human-wildlife conflicts that were thought to be intractable or with too high a welfare cost, but added: “That is not to say that individual welfare supersedes species or ecosystem welfare”.

In complex scenarios, featuring competing values, justified action may entail an unavoidable component of moral wrongdoing [37]. Finding the ‘least bad’ option may not be easy. There will frequently be trade-offs, for example between the rights of rural communities to safety from dangerous wildlife, and the well-being of individual animals. In these sometimes polarised debates both scientific evidence and competing values influence the process of finding acceptable solutions among diverse interest groups [38]. Supporters of a compassionate conservation agenda might reasonably argue that non-lethal options are sometimes not pursued for reasons of financial cost. The enormous cost of translocating elephants is, for example, one reason given for resorting to lethal methods [39]. Costs, including opportunity costs, are inevitably part of the consequentialist calculus. In this, the problem of identifying ‘right’ action has something in common with public health policy. The concept of the Quality Adjusted Life Year (QALY) is used in many countries to assess the benefit to human well-being and life expectancy against demands on a limited budget [40]. However, as Hayward et al. [6] pointed out, trade-offs such as these are not compatible with the extreme version of Compassionate Conservation advocated by Wallach et al. Oomen et al. drew attention to the potential for substantial costs to humans resulting from an extreme perspective on the moral salience of non-human sentience [31].

Securing populations, through carefully designed and humane interventions will, for numerous conservation problems, be justifiable, even where some individual animals suffer as a result. We reject Wallach et al.’s construal of ‘virtue’ in the hard cases where harm to individuals cannot be avoided in pursuit of a conservation goal, and they advocate taking no action. There are likely to be grave consequences of such inaction. An ethic of ‘stewardship’, acknowledging a duty to preserve biodiversity is enshrined in many national and international conservation agreements [12]. A recent survey of US respondents confirmed a duty to preserve wild animals for future people is widely supported [8]. Those future generations, looking back on continued biodiversity loss, may not agree that enduring ‘the full hurts of the world’ was an appropriate reaction to 21st century conservation challenges.

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
