# Peer review of "Consequences Matter: Compassion in Conservation Means Caring for Individuals, Populations and Species"

_animals, 2019, doi:10.3390/ani9121115_

Round 1
Reviewer 1 Report
This is a fine piece of work, delivered with authority by virtue of it collecting some of the more significant names in conservation in the global North. It is cogent, well argued, important and reasonable. It makes a valuable intervention into a debate which requires this sort of voice. Its arguments are clear, measured and right.
This could be published as is but I have five challenges for the authors on the MS itself and two about their more general publication strategy. The latter are pertinent because this intervention is not just an attempt to correct a debate that could go into blind alleys, but also an attempt to do so with authority. I infer this from the multitude of authors who have joined forces to put their name to a relatively short and simple essay. I think the authors will have to do this again, at other moments and in other journals, and if so then they will need to rethink a couple of aspects of their strategy.
On the paper
1.The paper needs more bite. Come up with a list of instances – in an table, not text, and drawn from your experience around the world (or better still, from the countries in which your authors are based) in which conservation goals will suffer from a compassionate approach. Done well this table would also include more difficult cases to consider (killing problem predators in Switzerland for example). But also lots and lots of more obvious examples. Gough island illustrates your point well, but I’d prefer you to hammer it home repeatedly with something extremely heavy.
Another possibility would be to come up with a list of conservation failures where doing nothing resulted in extinctions / system shifts etc.
2. And it may need more nuance – in particular more detail on the nature of the lethal methods – and especially debates about them – and how this is an area of anguish and concern in conservation, and how that has evolved over time. Some of the detail published on lethal methods in New Zealand a while ago may be a good place to start (Gillies et al 2003). It would be interesting to know if / how things have changed.
3. On style – you overstate the difficulty in the first paragraph. I do not think it is difficult to decide to kill stoats to protect juvenile kiwis. You are letting Wallach et al define the terms of the debate with this approach. Likewise start of para 3 is too weak. Lead with your strong points not your weak ones or your qualifications.
4. The paragraph sequence of section 4 did not work for me. It read as a list of different points rather than a coherent flowing argument. How does each paragraph follow on from the its predecessor?
5. The risk of perverse outcomes section was too short. And you need to point out the nonsense of doing nothing – it does not mean that nothing will happen. It denies the possibility that nature will exert its own agency and eg invasive rats will continue the destruction that they began. I would also reverse the orders of sections 3-5, because that way you are leading with your stronger points. You can be nice about areas of agreement at the end of the essay.
On the strateg
1. Your authorship is represents the global north well, and the global south badly. You weaken the force of your arguments with such a composition. I suspect that your networks are more far-reaching than this. I think you could improve content, broaden the geographical significance and resonance of your argument were you to decolonise your writing practices a bit more.
2. Be careful about sending such good material to an MDPI journal. The MDPI model is to set up multiple journals, bring in hundreds of editors and publish hundreds of special issues. Their freely available articles come at a smallish APC charge. As such it provides an apparently welcome change to the Elsevier style exploitation. But the business model hinges upon speed and volume. There is potential for this model to go wrong – and for the haste to result in mistakes being made by authors, reviewers or editors. My fears would be allayed were MDPI to publish the rejection rates for their journals. Rejection is bad for the business model. Every paper rejected is another APC lost. But I cannot see those data. Instead their website is full of information about how quickly they turn papers around. Please check this for yourselves – look at the variety of journals that they publish, the vast number of editors, and the huge number of special issues.
I can see why you are going to Animals – its quick, freely available and relatively cheap if you split the costs between so many authors. And because this looked like being an important intervention I agreed to review it. Normally I instantly refuse MDPI requests.
I mention this because it will pose two problems in the long run. First it increases the chances of your work being associated with a problematic article appearing in the same journal. Second, it means that you are publishing in a journal that is most likely going to be a continual mishmash of thinking. I do not see much sign of any intellectual coherence, agenda, brand or reputation emerging from all this volume of papers being processed. Again, please explore the special issues for yourselves and tell me if you think I am wrong about this. Journal management can be compared to flower arranging – carefully selecting the best blooms, rejecting many, and then pruning and improving the chosen few before putting them together. And by this analogy the MDPI approach is at the compost heap end of the flower arranging spectrum, whereas your work deserves a cut glass vase.
Please note I will gladly change my mind about MDPI were I to see data that shows that they are as selective and rigorous as they are fast and voluminous. My views are based on data available from their websites and I have brought this to your attention because I believe it is not widely known.
But otherwise many congratulations on such a good piece of work. I look forward to tweeting about this one.
References cited:
Gillies, C. A., Leach, M. R., Coad, N. B., Theobald, S. W., Campbell, J., Herbert, T., Graham, P. J. and Pierce, R. J. (2003) ‘Six years of intensive pest mammal control at Trounson Kauri Park, a Department of Conservation “mainland island”, June 1996 to July 2002’. New Zealand Journal of Zoology 30(4): 399–420
Author Response
Please see attachment. Contains response to reviewers one and two.

Reviewer 2 Report
I really appreciated this paper and the clarity that it provides in a difficult area. I am sure that there are plenty of additional examples that the authors are aware of and it is only through close familiarity that I draw their attention to the problems faced in disease suppression and rewilding experienced with the endangered Tasmanian devil. This also had to deal directly with the ethical/moral issues of the outcomes for individuals and the best outcomes for the population and ultimately the persistence of the species.
Author Response
See attachment - contains response to reviewers one and two.

Round 2
Reviewer 1 Report
An excellent paper well revised. Needs a quick proof check 'they promote' (line 109). Not doing the table as I recommended is fine but more signposting to the fact that Callen et al do that work for you would be better.
I must unsay my adverse remarks about the journal publishing house in the previous review. I have since investigated journal performance more thoroughly and realise my views were not well founded. I was wrong. My apologies to the editor for making them.